# Structural Studies of Pif1 Helicases from Thermophilic Bacteria

**DOI:** 10.3390/microorganisms11020479

**Published:** 2023-02-14

**Authors:** Stéphane Réty, Yingzi Zhang, Wentong Fu, Shan Wang, Wei-Fei Chen, Xu-Guang Xi

**Affiliations:** 1Laboratoire de Biologie et Modelisation de la Cellule, Ecole Normale Superieure de Lyon, CNRS, UMR 5239, Inserm, U1293, Universite Claude Bernard Lyon 1, 46 allee d’Italie, F-69364 Lyon, France; 2State Key Laboratory of Crop Stress Biology in Arid Areas, College of Life Sciences, Northwest A&F University, Xianyang 712100, China; 3Laboratoire de Biologie et Pharmacologie Appliquée (LBPA), CNRS, UMR 8113, ENS Paris-Saclay, Université Paris-Saclay, F-91190 Gif-sur-Yvette, France

**Keywords:** Pif1 helicase, X-ray crystallographic structure, SAXS, WYL domain, molecular modeling

## Abstract

Pif1 proteins are DNA helicases belonging to Superfamily 1, with 5′ to 3′ directionality. They are conserved from bacteria to human and have been shown to be particularly important in eukaryotes for replication and nuclear and mitochondrial genome stability. However, Pif1 functions in bacteria are less known. While most Pif1 from mesophilic bacteria consist of the helicase core with limited N-terminal and C-terminal extensions, some Pif1 from thermophilic bacteria exhibit a C-terminal WYL domain. We solved the crystal structures of Pif1 helicase cores from thermophilic bacteria *Deferribacter desulfuricans* and *Sulfurihydrogenibium *sp. in apo and nucleotide bound form. We show that the N-terminal part is important for ligand binding. The full-length Pif1 helicase was predicted based on the Alphafold algorithm and the nucleic acid binding on the Pif1 helicase core and the WYL domain was modelled based on known crystallographic structures. The model predicts that amino acids in the domains 1A, WYL, and linker between the Helicase core and WYL are important for nucleic acid binding. Therefore, the N-terminal and C-terminal extensions may be necessary to strengthen the binding of nucleic acid on these Pif1 helicases. This may be an adaptation to thermophilic conditions.

## 1. Introduction

Helicases are nucleic acids (NA) dependent nucleotide triphosphate (NTP) hydrolases. The chemical energy of NTP hydrolysis is converted into mechanical force to unwind double stranded NA (dsNA) into single stranded NA (ssNA) [1]. Helicases have been grouped in several Superfamilies characterized by conserved motifs [2]. Pif1 is a DNA helicase belonging to family SF1B of Superfamily 1 (SF1), which unwinds dsDNA with 5′ to 3′ directionality. Pif1 helicase has been identified in *S. cerevisiae* as a protein with multiple functions. Pif1 is involved in DNA repair and is necessary for the maintenance of mitochondrial and nuclear genomes integrity, in the regulation of telomerase activity and in the replication fork progression (reviewed in [3,4,5,6]). Many functions of Pif1 could be explained by its specific activity towards G-quadruplexes (G4), a particularly stable structure of NA which can be adopted by Guanine-rich sequences [7]. Although several helicases can unwind G4 [8], Pif1 exhibits the higher activity towards DNA G4 [9]. Pif1 is found in most organisms from bacteria to higher eukaryotes. In eukaryotes, Pif1 has two isoforms, located in nucleus and in mitochondrion. Pif1 and its related homologues Rrm3 and Pfh1 have been particularly studied in budding [10] and scission yeast [11] but functions of Pif1 in higher eukaryotes are less known. It appears that Pif1 functions in higher eukaryotes are more subtle than in yeast since mouse knocked–out for Pif1 are still viable [12] and exhibit only phenotypes related to myopathy [13]. However, Pif1 is related to cancer in humans since mutations in Pif1 have been linked to oncogenesis [14,15,16,17]. Sequences related to Pif1 were also found in some genomes, mostly of plants and bats, as being part of helitron transposons [18] and these Pif1-like sequences forming Hel domains of helitron transposases may have originated from bacteria [19].

Intriguingly, Pif1 helicases exhibit a high conservation in the helicase core (HC) domain from bacteria to humans. Since Pif1 from yeast and from humans have longer sequences and were difficult to purify, bacterial Pif1 helicases have been used as models to understand structure-function relationships for the whole family. Structures of Pif1 from *Bacteroides *sp. and *Thermus oshimai* were solved in complex with different ligand, such as nucleotides and ssDNA [20,21,22], replication fork mimetic substrate [23], and G4 [24]. Human Pif1 exhibits a high structural conservation with bacterial Pif1 [21,25], but budding yeast Pif1 helicases are even longer and exhibit an inserted extra domain [26,27]. Little is known about the physiological functions of Pif1 in bacteria, although many hypotheses have been proposed [28]. Pif1 is close to RecD helicase, which is involved in DNA repair when complexed in RecBCD [29]. Pif1 and RecD helicases may have overlapping functions in the repair of DNA double strand breaks and many bacteria such as *Escherichia coli* have a RecD copy but lack Pif1. Most thermophilic bacteria have one copy of Pif1 and it has been shown that some thermophilic bacteria, such as *Thermotoga elfii*, have an extra WYL domain at the C-terminal end of Pif1 [30], while other thermophilic bacteria, such as *Thermus oshimai*, do not have one [22]. Although dispensable, the role of this WYL domain may be important for thermophilic conditions.

In this work, we have studied Pif1 helicases from *Deferribacter desulfuricans*, and *Sulfurihydrogenibium *sp. which are thermophilic bacteria living at 65 °C [31,32]. Both possess a C-terminal WYL domain. For structural purposes, we studied only the HC domain truncated from WYL and solved the structures by X-ray crystallography in apo form and complexed to the nucleotide ADP or nucleotide analog AMPPNP. We showed by SAXS experiment that the N-terminal extension, which is necessary for ligand binding, is flexible in solution and we modelled the full-length protein, with the WYL domain, bound to ssDNA. The model reveals the potential importance of the N- and C-terminal parts for the regulation of nucleotide and nucleic acid binding.

## 2. Materials and Methods

### 2.1. Protein Expression and Purification

The helicase core domain from genes encoding *Deferribacter desulfuricans* Pif1 (Sequence ID: WP_013007012.1, DdPif1 residues 1-430) and *Sulfurihydrogenibium *sp. Pif1 (Sequence ID: HBT99145.1, SsPif1 residues 1-438) were cloned into plasmid pET15b-SUMO and then transformed into the C2566H *E. coli* strain (New England Biolabs, Ipswich, MA, USA). The expression and purification procedures are the same for both proteins. Briefly, when the culture reached the early stationary phase (OD_600_ = 0.55–0.6) at 37 °C, expression was induced at 18 °C over 16 h by the addition of 0.3 mM IPTG. Bacteria were harvested by centrifugation (4500× *g*, 4 °C, 15 min) and pellets were resuspended in lysis buffer (20 mM Tris-HCl pH 7.5, 500 mM NaCl, 10 mM Imidazole, and 5% glycerol (*v*/*v*)). Bacteria were broken with a French press and then further sonicated 2–3 times to shear DNA. After centrifugation at 12,000 rpm for 40 min, the supernatants were filtered through a 0.45-μm filter and loaded onto a Ni^2+^ charged IMAC column (Cytiva). After washing twice, the SUMO-Pif1 was then eluted from the Ni^2+^ affinity column with elution buffer (20 mM Tris-HCl, pH 7.5, 500 mM NaCl, 300 mM Imidazole, and 5% glycerol (*v*/*v*)) at 4 °C. The eluted protein was treated with SUMO protease (Invitrogen, Beijing, China) and the digested protein was further purified by a HiTrap Heparin column (Cytiva) to remove the SUMO-tag and other protein impurities. The eluted fraction containing Pif1 was collected and concentrated. The final purified protein was dialyzed against the storage buffer (20 mM Tris-HCl, pH 7.5, 500 mM NaCl, 1 mM DTT) and concentrated to approximately 10 mg/mL for crystallization trials. Purity was assessed by SDS-PAGE and was determined as 95%.

### 2.2. Crystallization of Pif1 Proteins

Crystallization trials for DdPif1 apo, DdPif1-AMPPNP, and SsPif1-ADP were performed at 20 °C by the sitting-drop vapor diffusion method. Proteins were mixed with the nucleotide (AMPPNP or ADP) prior to crystallization at the molar ratio of 1:1.5. The initial DdPif1 apo crystals were obtained from the precipitant solution containing 0.12 M (NH_4_)_3_Cit, 15% (*w*/*v*) PEG MME 2000 (polyethylene glycol monomethyl ether 2000). This condition was optimized by a grid search by using 24-well Linbro plates at 20 °C where 1 μL of protein sample and 1 μL of precipitant were mixed together and equilibrated with 0.4 mL of precipitant. Crystals of the DdPif1-AMPPNP complex were obtained from the precipitant containing 0.1M Bis-Tris propane (pH 6.5), 0.1 M calcium acetate, and 10–15% PEG4000, and crystals of the SsPif1-ADP complex were obtained from the precipitant containing 0.1M Hepes (pH 7.5), 3.5–3.6 M sodium formate, 5% glycerol, and 0.01 M spermidine.

### 2.3. Data Collection, Model Building and Structure Refinement

X-ray diffraction data were collected at beamline BL17U1 at Shanghai Synchrotron Radiation Facility (SSRF). Data were processed with AUTOPROC software [33] with XDS [34] for indexing and data reduction and AIMLESS [35] for scaling. All structures were solved by molecular replacement using the BsPif1 apo structure (PDB 5FTD) as the search model using the PHASER module in PHENIX software [36]. The models were manually adjusted with COOT [37] and refined in PHENIX, including twin law when necessary. The data and refinement statistics are summarized in Appendix A.

### 2.4. Small-Angle X-ray Scattering

The scattering data of apo DdPif1 were collected at beamline BL19U2 of SSRF in batch mode at 25 °C and at protein concentrations 1.0, 3.0, 5.0, 7.0, and 9.0 mg/mL in buffer containing 25 mM Hepes (pH 7.5), 100 mM NaCl, and 5% Glycerol. The wavelength (λ) of X-ray radiation was set as 1.033 Å, and a sample to detector distance was set to 2.5 m. The X-ray beam with a size of 0.40 × 0.15 (H × V) mm^2^ was adjusted to pass through the centers of the capillaries for each experiment. The data were processed with BioXtasRAW [38]. Then, 20 consecutive frames of 1 s exposure time recorded for each sample were averaged after checking for absence of radiation-induced protein damage. The scattering patterns of the corresponding buffer solution were recorded before and after the measurements of the protein sample, and the averaged buffer pattern was subtracted from the protein patterns. The radius of gyration (Rg) and maximum particle dimension (Dmax) were extracted using BioXtasRAW and GNOM program from ATSAS suite [39]. Ab initio envelopes were determined using 50 models calculated either by DAMMIF [40] or by DENSS [41], with experimental Rg and Dmax values as constraints. The quality and uniqueness of the results were assessed further by averaging with DAMAVER for DAMMIF models and with the internal procedure of DENSS. The agreement between the replicated models was quantified by the mean normalized spatial discrepancy (NSD). The atomic model of apo DdPif1 was generated by flexible modelling using DADIMODO [42] where flexibility was introduced in the N-terminal tail and between domains 1A and 2A. Atomic models were fitted in ab initio envelopes with the Supcomb algorithm [43]. A comparison of the scattering of the resulting all-atom model with experimental data was computed with CRYSOL. SAXS data collection and analysis were summarized in Appendix A.

### 2.5. Modeling and Molecular Dynamics Simulations

As DdPif1 structure is not available in the Alphafold database, the full-length DdPif1 structure (residues 1-510) was modelled using Alphafold [44] and RosettaFold [45] installed on a local computer. From the Alphafold model, domains 2A and 2B were manually moved in order to mimic the conformation of BsPif1 in complex with ssDNA (PDB 5FTE). ssDNA was built in the model using the path provided by the structure 5FTE inside the helicase core and the binding mode of ssDNA to WYL the domain in the structure of the *C. crescentus* DriD C-domain (PDB 7U02). The ATP nucleotide was modelled using DdPif1-AMPPNP complex information. The DdPif1-ATP-ssDNA was then subjected to a molecular dynamics simulation. The inputs were prepared using CHARMM-GUI [46] with the Solution Builder module. The CHARMM36m force field was used to describe the full system consisting of a box with explicit water and KCl (reaching 150 mM). The CHARMM-GUI protocol was followed using GROMACS (Version 2021.4) [47]. The NVT equilibration phase was performed during 50 ps. The trajectory of 50 ns was obtained and further analyzed with the GROMACS tools and visualized in PyMOL 2.4 [48], which served to create all of the illustrations.

## 3. Results

### 3.1. Structure Determination of DdPif1 and SsPif1

In an initial effort to crystallize Pif1 from *Homo sapiens* or *Saccharomyces cerevisiae*, we screened for crystallization assays several recombinant Pif1 produced from different bacterial species, including thermophilic bacteria. We were able to crystallize the helicase core of Pif1 from the thermophilic bacteria *Deferribacter desulfuricans* and *Sulfurihydrogenibium *sp., which are moderate thermophilic bacteria, living at an average temperature of 65 °C. Pif1 from these bacteria are named DdPif1 and SsPif1, respectively. DdPif1 and SsPif1 consist of a helicase core (HC) domain, with 45.0 and 37.6 % identity with *Bacteroides *sp. Pif1 (BsPif1), respectively, and a WYL domain in the C-terminal position (Figure 1A). For crystallization, we therefore used constructs of the HC of DdPif1 (aa 1-433) and SsPif1 (aa 1-438), which are homologous to the HC of *Homo sapiens* Pif1 (HsPif1).

The structures were solved by molecular replacement (MR) using the apo structure of BsPif1 (PDB 5FTD). DdPif1 apo crystallized in spacegroup C222_1_ with one molecule in the asymmetric unit and DdPif1 in the complex with the nucleotide analogue AMPPNP crystallized in spacegroup P2_1_2_1_2 with two molecules in the asymmetric unit (Appendix A). Though solving this structure was not particularly difficult because of strong homology with the template structure BsPif1, used as a search model for MR, the DdPif1-AMPPNP structure has a possible domain swapping between the two molecules of the asymmetric unit. Indeed, the C-terminal half of the two Pif1 molecules, containing domains 2A and 2B, can be exchanged in a region where the electron density is ambiguous. There are therefore two possibilities for describing the Pif1 structure (Appendix A). However, we choose to consider the two molecules folded as compact structures with no swapped domains, since domain swapping of RecA domains in helicases was never reported before.

SsPif1 in complex with ADP has an apparent spacegroup C222_1_, with unit cell a = 141.28 Å b = 178.892 Å c = 129.827 Å α = β = γ = 90 °C containing one molecule but no solution was found with MR using PDB 5FTD as a search model. By testing different subgroups, it was found that indexing in spacegroup P2_1_ with unit cell a = 113.138 Å b = 129.29Å c = 113.68 Å α = 90 °C β = 103.4 °C γ = 90 °C gave an acceptable MR solution. This spacegroup harbors a strong translational non-crystallographic symmetry (tNCS) and twinning with twin law l, -k, h. An unambiguous solution was found in MR with four molecules in the asymmetric unit. The four molecules are arranged in two dimers related by tNCS. In the dimer, the molecules are related by a non-crystallographic two-fold symmetry, interacting by a β-strand in the 2B domain (Appendix A).

### 3.2. Overall Structure of DdPif1 and SsPif1

The helicase core of Pif1 from thermophilic bacteria is very similar to other bacterial Pif1 structures already solved (Figure 2). The HC contains the features of previously described SF1B helicases: two RecA-like domains 1A and 2A containing insertions 1B and 2B, respectively (Figure 1A). Conserved motifs are present in DdPif1 and SsPif1 domains with motifs I, IA, II, and III in domain 1A and motifs IV, V, and VI in domain 2A (Figure 1B). Domains 1A and 2A have the characteristics α/β fold of the RecA domains with 1A folded as a parallel β-sheet with five β-strands flanked by three and four α-helices at each side and 2A folded as a parallel β-sheet with three β-strands surrounded by seven α-helices. The 2B domain is folded as the SH3 domain, as shown in RecD2 from *Deinococcus radiodurans* (PDB ID 3GPL) and Dda from T4 bacteriophage (PDB ID 3UPU), with a long β-hairpin protruding, named loop-3 in BsPif1 [20]. Domain 1B has a less conserved structure, as it has been shown to be folded as an α-helix or a loop. Domain 1B has been proposed to act as a wedge during dsDNA unwinding.

When DdPif1 apo is superimposed on BsPif1 apo (PDB 5FTD) (Figure 2F), BsPif1 (PDB 5FHG), or ToPif1 (PDB 6S3E), the root mean square deviation (rmsd) on 360 Cα is 3.34, 3.58, and 3.46 Å, respectively. Conversely, Cα rmsd for SsPif1 superimposed on the bacterial structure complexed with the nucleotide, but no ssDNA BsPif1-ADP (PDB 5FTC), BsPif1-ADP-AlF (PDB 5FHF), and DdPif1-AMPPNP is 2.32, 2.26, and 2.71 Å, respectively, over 416 residues (Figure 2E,F).

Only DdPif1 was solved in two different states, apo and in complex with the ATP analogue AMPPNP (Figure 2A,B, and Figure 3A,B). Structural changes upon nucleotide binding are moderate in the absence of ssDNA (Figure 2C), as already observed for BsPif1 and ToPif1. The main difference is the folding of the N-terminal part of DdPif1 (residues 1-20). This N-terminal part is not visible in the DdPif1 structure and may be disordered (Figure 2A). However, when AMPPNP is bound, the N-terminal part is visible and contributes to nucleotide binding. The same feature is observed in the SsPif1-ADP structure where the N-terminal part (residues 1-22) participates in ADP binding (Figure 2D).

### 3.3. Comparison of Nucleotide-Binding Site

Only one molecule of DdPif1 in the asymmetric unit is complexed with AMPPNP (Figure 3B). The nucleotide has a good electron density which was also checked with the calculation of a simulated annealing omit map (Appendix A). Since the resolution of these structures is limited to 3.2 Å, details of interaction, in particular Mg^2+^ and water molecules, could not be built. However, the comparison with the previous structure of Pif1 solved with a nucleotide at a higher resolution shows that the nucleotide binding site is conserved (Figure 3B–D). The nucleotide binding pocket sits between domains 1A and 2A but most of the interactions with the base involve only domain 1A and the N-terminal tail. The adenine base is stacked by an aromatic residue of motif IV (Y183, Y186 or F187 in DdPif1, SsPif1, and BsPif1, respectively) and a hydrophobic site chain of a residue of the N-terminal tail (A4, V5 or M4 in DdPif1, SsPif1, and BsPif1, respectively). Several residues from Motif I are implicated in the coordination of α- and β-phosphate moiety (Figure 2A). Most of these residues are interacting through their peptidic bond, K28 to 735 for DdPif1 and A31 to T37 for SsPif1.

### 3.4. Analysis of DdPif1-Apo Structure in Solution by SAXS

Since we observed that the N-terminal part is not visible in the DdPif1 apo structure and contributes to nucleotide binding, we hypothesized that it could flexible. Therefore, we analysed the structural changes of DdPif1 apo in solution by small-angles X-ray scattering (SAXS). The molecular weight of DdPif1, estimated from I (0), Porod volume, and Volume of correlation (Vc) is 50 kD, which is consistent with a monomer of DdPif1 in solution. If the N-terminal part is added to the model of DdPif1 apo with the conformation of the DdPif1-AMPPNP complex and fitted to the SAXS data, the χ^2^ is 10.1. This high χ^2^ value shows that the N-terminal part in solution is not in the conformation of the DdPif1-nucleotide complex. Thus, the DdPif1-apo structure was modelled by flexible fitting with DADIMODO by introducing flexibility in the N-terminal tail (residues 1-20) and the linker between 1A and 2A domains. The fitted structure has a reduced χ^2^ of 1.22, which exhibits a good fit with the SAXS data (Figure 4A). The Krakty plot indicates that the DdPif1-apo structure is mainly globular (Figure 4B). The DdPif1 apo model could be fitted in the SAXS envelope computed ab initio by DAMMIF (Figure 4C) of DENSS (Figure 4D) algorithms. Ab initio modelling approaches, with a resolution ranging from 25 to 31 Å, show two parts: a globular region where the helicase core can be fitted, and an elongated volume occupied by the flexible N-terminal part. Therefore, in the absence of a nucleotide, the N-terminal may be unstructured and becomes folded upon nucleotide binding.

### 3.5. Modelling of the Structure of the BsPif1-ssDNA Complex

The full length DdPi1 structure (residues 1-510) was predicted with no ligand by Alphafold and RosettaFold algorithms, both giving very similar results with differences in domains positions (rmsd 5.7 Å). In order to get a model of DdPif1 bound to ssDNA, we manually moved the domains 2A and 2B in order to fit ssDNA as in known structures of Pif1-ssDNA complexes, such as in BsPif1-ssDNA complex (PDB 5FHD). Alphafold predicts the fold of a WYL domain in the C-terminal part of DdPif1. The WYL domain has been shown in TePif1 to be crucial for helicase activity and was shown to bind to the 5′ end of ssDNA [30]. In the Alphafold model, the WYL structure is similar to the solved WYL domain, in particular to the structure of the *C. crescentus* DriD C-domain bound to ssDNA (PDB 7U02) (Figure 5A and Appendix A). We used this information to model a ssDNA fragment bound to WYL and we joined it to the ssDNA bound to HC. The final modelled ssDNA contains 18 nucleotides with the 5′ end bound to WYL and the 3′ end bound to HC. In order to remove clashes introduced during the manual adjustment of domain positions and the building of ssDNA, the structure was minimized and relaxed with a 50 ns molecular dynamics simulation. The final model is shown in Figure 6.

In the DdPif1-ssDNA model, ssDNA is extended from the WYL domain (5′ end) to the end of nucleic acid binding channel of HC (3′ end). The DNA duplex, which could be unwound, would be attached to the 3′ end and would be near the 1B and 2B domains, which have been shown to be necessary for helicase activity [20]. The most interesting feature is the detail of the binding of ssDNA from the exit tunnel of HC to the WYL domain. ssDNA is closely in interaction with the domain 1A, DC2 motif, and WYL domain. Several positively charged residues may be important for the interaction with the ssDNA ribose-phosphate backbone, in particular K151 in domain 1A, K413 and K414 in DC2, and R464, R491, R495, R498, and R501 in the WYL domain. Most of the residues highlighted in the WYL domain were already described in TePif1 (R470, R501, R504) and were later confirmed in structural studies of WYL containing proteins to be important for ligand binding in DriD [50] and PafB/PafC [51]. The predicted fold of the DdPif1 WYL domain is similar to the DriD WYL domain with rmsd of 2.64 Å over 72 Cα. In particular, residues that are important for the ssDNA binding of ssDNA in the DriD WYL domain (Figure 5A) are conserved in the DdPif1 WYL domain (Figure 5B), but aromatic residues Y168 and Y192 in DriD are not conserved in DdPif1. The WYL domain contributes extensively to binding and maintaining ssDNA in an extended conformation.

## 4. Discussion

Crystal structures of DdPif1 and SsPif1 confirm the high level of conservation among Pif1 helicase family of the HC domain and the specific features of the 2B domain fold with its extended β-hairpin, named loop-3 in BsPif1 [20]. The nucleotide binding mode on the 1A domain is conserved and involves the N-terminal tail which interacts with the base by a hydrophobic amino acid (A4, V5, M4 in DdPif1, SsPiF1 and BsPif1, respectively) and an aromatic residue at the end of 1A domain (Y183, Y186, F187 in DdPif1, SsPiF1 and BsPif1, respectively). When no nucleotide is present, the N-terminal tail is not completely folded, as seen by the SAXS experiment in solution. Though the HC is conserved, there exist some structural differences in separation wedge regions introducing divergence of the unwinding mechanism among bacteria, yeast, and human Pif1 [25]. Furthermore, the Pif1 family has evolved by the lengthening of the N-terminal and C-terminal tails and the addition of accessory domains [52]. These extra domains may be important for helicase activity, recruitment, regulation, or even for conferring novel functions to Pif1 proteins.

DdPif1 and SsPif1 share a high homology with TePif1, in particular, by the presence of the C-terminal WYL domain. WYL domains were initially identified in prokaryotic proteins involved in the defense system, in particular, in the CRISPR-Cas loci [53], but were found in many other proteins. Recently, the structure of several proteins containing WYL domains were solved, showing diverse functions of these domains as modules for protein-protein interaction [54], ligand binding modulator [55], and binding of ssDNA [50]. Here we modeled the WYL domain in Pif1 helicase. Though we have not conducted a biochemical analysis of DdPif1 and SsPif1, we can expect a similar behavior as TePif1, which was extensively characterized. In particular, WYL was shown by deletion and point mutation analysis to be crucial for ssDNA binding and helicase activity [30]. Our model generated from an Alphafold prediction shows that the C-terminal part containing UvrD-C-2 domain and WYL is involved in ssDNA binding. The residues predicted in TePif1 to be important for ssDNA binding were confirmed by the structure of the WYL domain complexed to ssDNA [50]. The DdPif1 WYL model reproduces the same features, highlighting the conserved positively charged residues as important for ligand binding. Since the WYL domain binds to the 5′ end of ssDNA, it may be important for initiating the binding of ssDNA by Pif1 and tightening the ssDNA in the HC channel.

Recently, the conformations of ssDNA outside of the Pif1 HC tunnel have been explored by molecular dynamics and magnetic tweezers [56]. A stretched ssDNA conformation reduces translocase and helicase activity, demonstrating that ssDNA have to interact with Pif1 outside of the HC channel. Though a part of ssDNA is visible on the crystal structure of Pif1-ssDNA complexes, it has thus been hypothesized that Pif1 helicase has other ssDNA binding sites at its surface which could interact with ssDNA and may regulate the helicase activity. In particular, the 3′ end of unwound ssDNA may also interact with Pif1 [57], which could explain the strand-switching behavior of Pif1 when a G4 roadblock is encountered [58]. The WYL domain offers a larger binding surface for ssDNA and may regulate functions associated with the other binding sites at the surface of Pif1.

However, the WYL domain is not always present in Pif1, even in thermophilic bacteria, since ToPif1 for instance does not contain one. Furthermore, the role of Pif1 helicase in thermophilic bacteria is still obscure. Pif1 has an important activity towards G4 but the frequency of predicted G4 presence is not correlated with high GC% content [59,60]. For instance, *Sulfurihydrogenibium *sp. belongs to the phyla Aquificae which does not exhibit a genome with high GC% content and has a low frequency of G4. The situation is the same for *Thermogoga elfii*, which belongs to phyla Thermotogae and which has lower G4 frequency than expected by chance [59].

How does Pif1 helicase in thermophilic bacteria achieve thermostability? Several observations concerning protein thermostability have been done, regarding salt bridges, amino acid composition, and entropic stabilization [61,62,63]. It has been hypothesized that amino acid composition could be a signature of thermophilic character [64] but the comparison of the sequence of Pif1 from mesophilic and thermophilic bacteria does not exhibit any significant difference. Since the growth temperature of *Deferribacter desulfuricans* and *Sulfurihydrogenibium *sp. is around 65 °C, Pif1 activity should be tested at these temperature but is experimentally difficult, as discussed for TePif1 [30]. Therefore a specific setup should be developed, as for the study of MCM helicase from archaea [65]. More biochemical studies are thus required to establish the structure-function relationships of thermophilic Pif1 helicases.

## Figures and Tables

**Figure 1 microorganisms-11-00479-f001:**
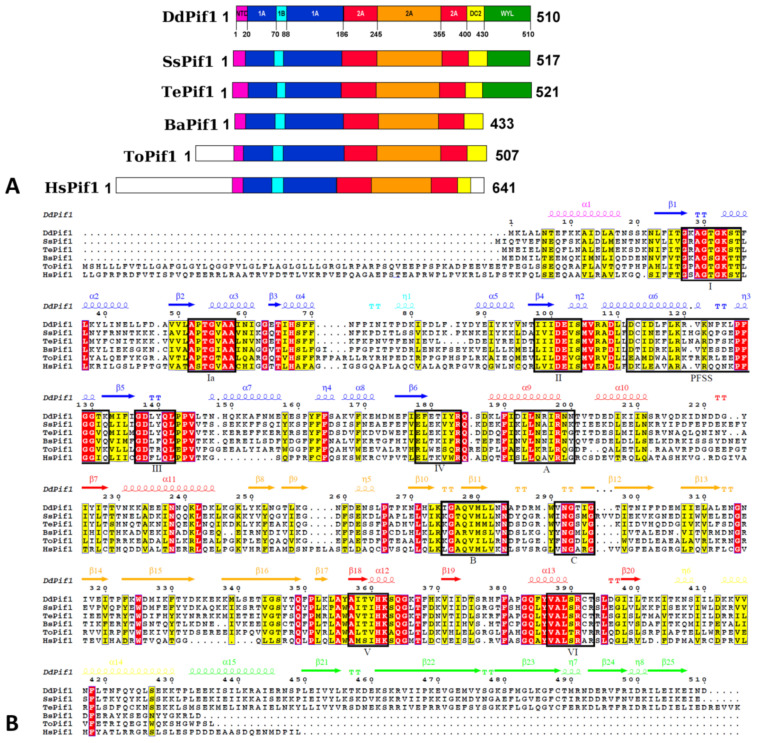
(**A**) Domains organization of Pif1 molecules from *Deferribacter desulfuricans* (DdPif1)), *Sulfurihydrogenibium* sp. (SsPif1), *Thermotoga elfii* (TePif1), *Bacteroides* sp. (BsPif1), *Thermus oshimai* (ToPif1), and *Homo sapiens* (HsPif1). Domains are colored as N terminal tail (magenta), domain 1A (blue), domain 1B (cyan), domain 2A (red), domain 2B (orange), UvrD_C_2 domain (DC2) (yellow), WYL domain (green). Non homologous N-terminus are kept white. (**B**) Sequence alignment of Pif1 helicases with highlighted homology conservation and motifs (HsPif1 is truncated from resides 1-138). The secondary structure of DdPif1, based on 3D structure, is shown colored as domains. Figure prepared with Espript [49].

**Figure 2 microorganisms-11-00479-f002:**
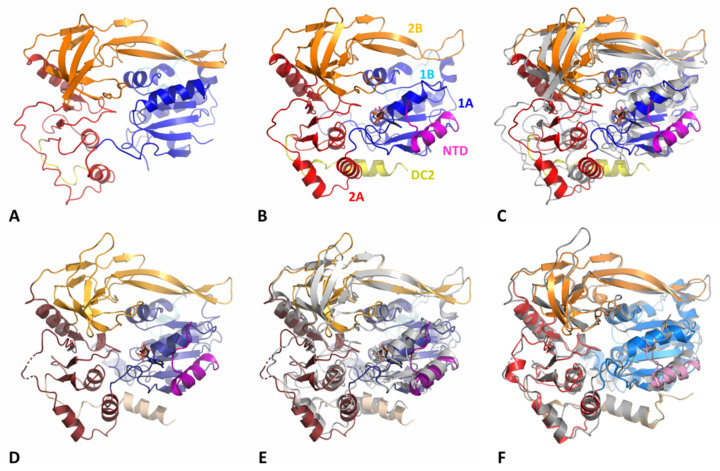
Structures of bacterial Pif1. The structures are shown in the same orientation and have been superimposed on domain 1A. (**A**) DdPif1 apo. (**B**) DdPif1-AMPPNP. (**C**) Superimposition of DdPif1-AMPPNP on DdPif1 apo (grey). (**D**) SsPif1-ADP. (**E**) Superimposition of SsPif1 on DdPif1-AMPPNP (grey). (**F**) Superimposition of BsPif1-ADPAlF_4_ complex (PDB 5FTE) on SsPif1-ADP (grey). Domains are colored, as in Figure 1A, and nucleotide is shown as sticks.

**Figure 3 microorganisms-11-00479-f003:**
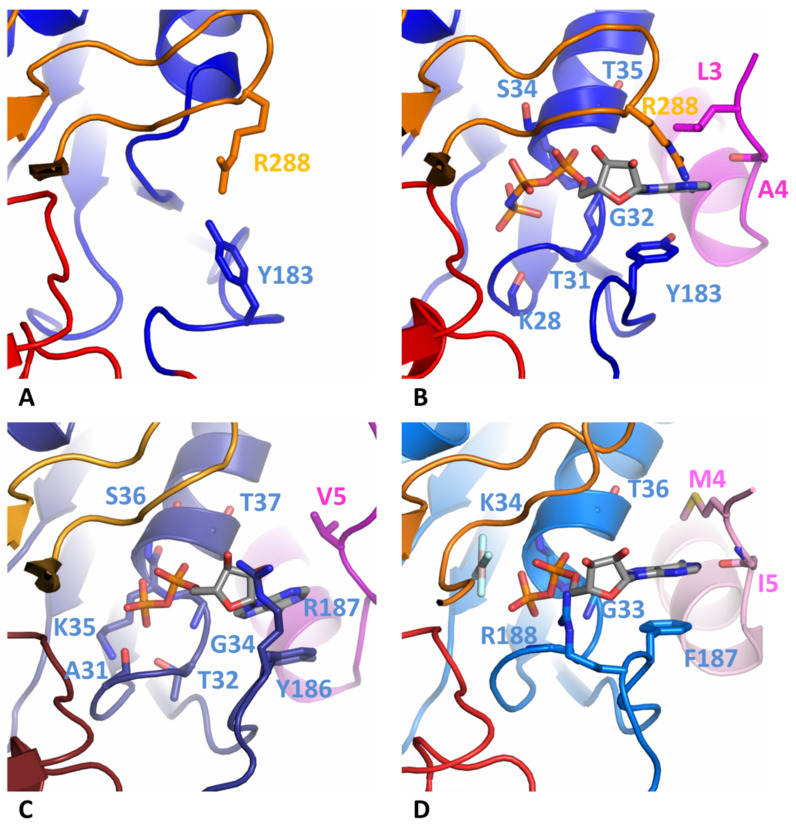
Nucleotide binding site. (**A**) DdPif1 apo. (**B**) DdPif1-AMPPNP. (**C**) SsPif1-ADP. (**D**) BsPif1-ADP-AlF_4_ (PDB 5FHF). Nucleotide is colored grey and important residues are shown as ball-and-sticks.

**Figure 4 microorganisms-11-00479-f004:**
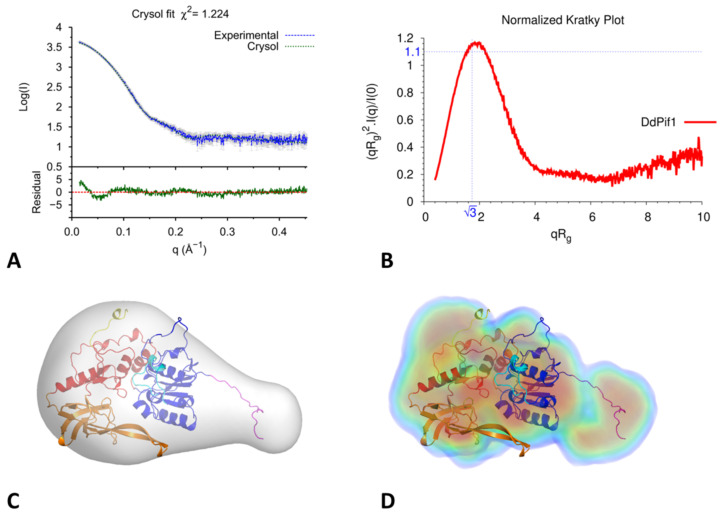
SAXS analysis of DdPif1 apo. (**A**) Computed SAXS curve from the DdPif1 apo model in solution, fitted to experimental SAXS curve of DdPif1 (blue). The residual between calculated and measured intensities is shown below. (**B**) Normalized Krakty plot of DdPif1-apo (the point (√3,1.1) corresponds to a perfect sphere). Ab initio envelope of DdPif1 apo calculated by DAMMIF (**C**) and DENSS (**D**).

**Figure 5 microorganisms-11-00479-f005:**
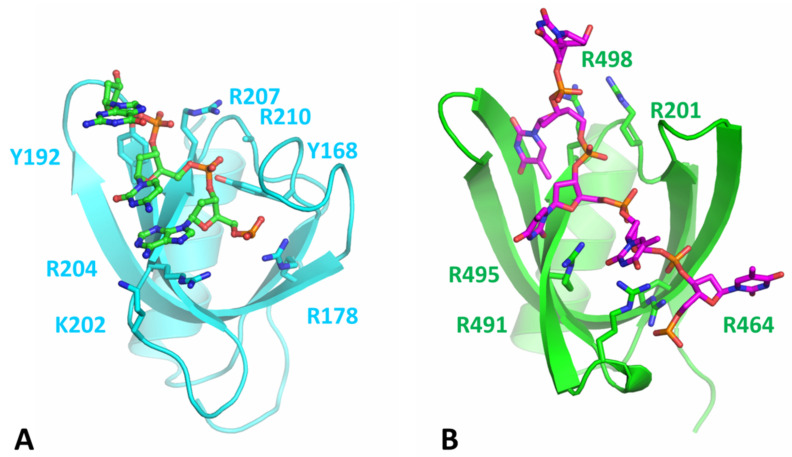
Modeling of the DdPif1 WYL domain bound to ssDNA. (**A**) DriD WYL domain bound to ssDNA (green). Important residues for ligand binding are highlighted as in [50] (side chain of R210 is not present in structure 7U02). (**B**) Prediction of the WYL domain of DdPif1 with ssDNA (magenta) and identified important residues for binding. The structures are shown in the same orientation.

**Figure 6 microorganisms-11-00479-f006:**
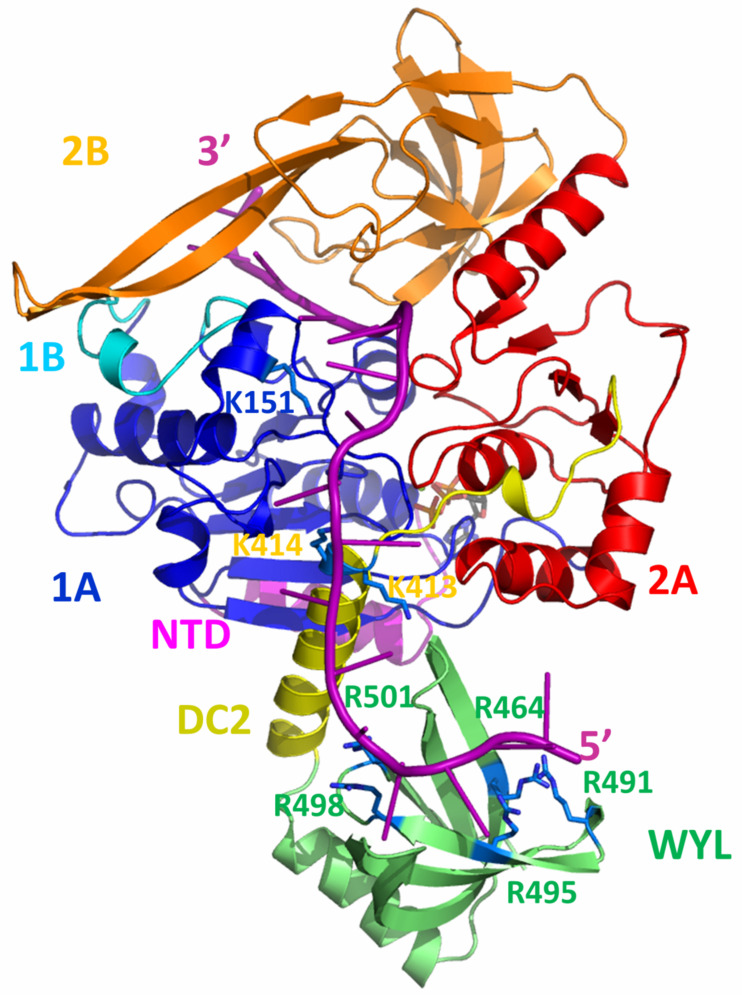
Model of the full-length DdPif1 bound to ssDNA. Domains are colored as in Figure 1A and 18-mer ssDNA is shown in magenta. Important residues involved in ssDNA binding are shown in blue as sticks.

## Data Availability

The structures have been deposited at the PDB with the codes 8BNV, 8BNX, and 8BNS for DdPif1 apo, DdPi1-AMPPNP, and SsPif1-ADP, respectively.

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
