# Peer review of "Structural Studies of Pif1 Helicases from Thermophilic Bacteria"

_microorganisms, 2023, doi:10.3390/microorganisms11020479_

Round 1

Reviewer 1 Report

Pif1 is a conserved helicase from bacteria to humans and has the particularity to be involved in many functions and to act on many different substrates, which makes it particularly interesting to study. The authors have acquired great expertise in the structure-function analysis of yeast and thermophilic bacteria Pif1, and it is interesting to be able to benefit from their work on other members of the bacterial system harboring the WYL extra-domain, which is less well documented.

The reconstruction of the whole flexible protein in complex with the ssDNA is elegantly conducted using the available structural, dynamics, and modeling techniques, and the adequate structural models available in the PDB. Overall the article, built on structural data only, is well constructed, but it lacks a thorough proofreading before being published.

I have so only a few minor remarks:

-          Although the color code of the domains is well explained in Fig1A and reported on all the structures, it would be useful to add some indications on the figures in order to help the reader (indicate where are the Nter and Cter, annotate the domains etc…)

-          The main text and legends deserve a thorough re-reading to eliminate the many small errors (examples not exhaustive: “the” is almost systematically missing; line22: domain 1A; line24: these Pif1 helicases….). I am not bilingual but it seems to me that English deserves to be improved

-          the last sentence of the introduction may be over-interpreting the data and should be changed

-          Title of paragraph 2.2: remove “Purification”

-          Line 116: remove “model”

-          Line 121: add “protein” concentrations

-          Line 138: “In” N-terminal tail and “atomic” models

-          Line 139: give a reference for "Supcomb algorithm” ?

-          3.1 : choose between "space group" and "spacegroup”

-          Figure 1A : thanks for these color codes, but the domain names are not readable on the dark rectangles

-          The figure 1B is not of sufficient quality to be readable, and cite the software that allowed it to be built

-          legend to figure 2: nucleotide in sticks ?

-          3.3 : Figure 3C is not cited in the text

-          Figure 3C : although the resolution is only 3.1A, is it not possible to specify the bonds (H or other?) between the residues and the nucleotides ?

-          Line 277 fig4 legend : “Ab initio”

-          many names of bacteria are not in italics

-          line 300: “in interaction with”

-          line 322 : Crystal “structures”

Author Response

We thank the reviewer for the careful reading of our manuscript and the positive comments. We addressed all the points highlighted.

- Although the color code of the domains is well explained in Fig1A and reported on all the structures, it would be useful to add some indications on the figures in order to help the reader (indicate where are the Nter and Cter, annotate the domains etc…)

Answer:

We have added annotations of N-ter and Cter on the figures 2 and 6

- Line 139: give a reference for "Supcomb algorithm” ?

Answer:

We have added a reference for Supcomb

- Figure 1A : thanks for these color codes, but the domain names are not readable on the dark rectangles

The figure 1B is not of sufficient quality to be readable, and cite the software that allowed it to be built

Answer:

Figure 1A was changed with white labels in dark rectangles. We added a reference for the software used for Figure 1B. The Figure 1 has been uploaded with high resolution (19x19cm, 300dpi)

The last sentence of the introduction may be over-interpreting the data and should be changed

Answer:

We have modified this last sentence in order to be less peremptory and more hypothetic

- Figure 3C : although the resolution is only 3.1A, is it not possible to specify the bonds (H or other?) between the residues and the nucleotides ?

Answer:

Interactions between protein, nucleotide and Mg2+ ion may involve water molecules which are not visible at this resolution (3.1Å). Furthermore, side chains and rotamers are not clearly defined. Therefore it would be difficult to give an atomic description of the binding of nucleotide without over-interpreting the structure.

- The main text and legends deserve a thorough re-reading to eliminate the many small errors (examples not exhaustive: “the” is almost systematically missing; line22: domain 1A; line24: these Pif1 helicases….). I am not bilingual but it seems to me that English deserves to be improved

Title of paragraph 2.2: remove “Purification”

Line 116: remove “model”

Line 121: add “protein” concentrations

Line 138: “In” N-terminal tail and “atomic” models

choose between "space group" and "spacegroup”

legend to figure 2: nucleotide in sticks ?

Figure 3C is not cited in the text

Line 277 fig4 legend : “Ab initio”

many names of bacteria are not in italics

line 300: “in interaction with”

line 322 : Crystal “structures”

Answer:

We have corrected the text accordingly. We have also carefully checked the text for misspelling and corrected English style

Reviewer 2 Report

The manuscript by Réty et al. presents the crystal structures of Pif1 helicase core domains from two thermophilic species Deferribacter desulfuricans (Db) and Sulfurihydrogenibium sp (Ss) in apo (DbPif1) and nucleotide (ATP analogs) bound (DbPif1 and SsPif1) forms. Interestingly, while the core domain of Pif1 is conserved from bacteria to humans, some thermophilic bacteria such as Db and Ss have an additional C-terminal WYL domain. The function of WYL domain is not well understood and could range from second messenger binding in proteins associated with CRISP Cas genes (Ref 51) to single-strand (ss) DNA binding for allosteric regulation in transcription (Ref 48).

As expected, the overall structures of the helicase core domains of DbPif1 and SsPif1 are similar to the previously described structures of Pif1 from other species.  Because the structures in this study are solved to medium resolution (2.86-3.12 Å for DbPif1 and 3.24 Å for SsPif1) the fine details of interactions with the nucleotide cannot be established, but the authors show that in general they are very similar to the ones described in homologous higher-resolution structures. The authors employ SAXS to further examine the conformation of the apo form of DbPif1 since the N-terminal   residues (1-20) of DbPif1 in the apo structure are disordered but are ordered in the nucleotide bound structure and interact with the nucleotide. They confirm that the conformation of apo DbPif1 in solution resembles the one observed in crystal and is globular and monomeric; the positioning of the N-terminal residues is flexible but could be accommodated by the elongated extension.  Furthermore, the authors model DbPif1 in complex with ssDNA on a basis of a known helicase-DNA structure and show that WYL domain might participate in DNA binding whereby suggesting its role if the domain is associated with a helicase.

Overall, the structural studies are technically well performed and described and the conclusions of the study appear to be valid.

I have one concern:

The authors state (lines 176-181) that “the DdPif1-AMPPNP structure has a possible domain swapping between the two molecules of the asymmetric unit. Indeed the C-terminal half of the two Pif1 molecules, containing domains 2A and 2B, can be exchanged in a region where the electron density is ambiguous. There are therefore two possibilities for describing Pif1 structure.. However we choose to consider the two molecules folded as compact structures with no swapped domains.”

What is the evidence for the “compact” conformation of the DdPif1-AMPPNP structure? Are there any homologous proteins observed in a “domain swapped” conformation when bound to AMPPNP? How many residues of the linker region between domains 2A and 2B are disordered? Can SAXS that does show a compact globular conformation for apo DbPif1 be applied to confirm the compact nucleotide bound conformation?

Author Response

We thank the reviewer for the careful reading of the manuscript and the positive comments.

The authors state (lines 176-181) that “the DdPif1-AMPPNP structure has a possible domain swapping between the two molecules of the asymmetric unit. Indeed the C-terminal half of the two Pif1 molecules, containing domains 2A and 2B, can be exchanged in a region where the electron density is ambiguous. There are therefore two possibilities for describing Pif1 structure.. However we choose to consider the two molecules folded as compact structures with no swapped domains.”

What is the evidence for the “compact” conformation of the DdPif1-AMPPNP structure? Are there any homologous proteins observed in a “domain swapped” conformation when bound to AMPPNP? How many residues of the linker region between domains 2A and 2B are disordered? Can SAXS that does show a compact globular conformation for apo DbPif1 be applied to confirm the compact nucleotide bound conformation?

Answer:

We prefer to describe the structure as compact rather than domain swapped because there is no example in the literature of dimers of helicases with such a domain swapping. Though many helicases have been shown to form dimers upon DNA binding (SF1 helicase as UvrD from Escherichia coli or Pif1 from Saccharomyces cerevisiae), none have been describe with an exchange of RecA domains. In our case, the domain swapping involves a mandatory dimerization of Pif1 upon nucleotide binding, with no DNA. However we do not observe a dimerization of DdPif1 upon ADP or AMPNP binding. We observed a dimerization of ToPif1 in presence of a long ssDNA (Dai et al, doi: 10.1093/nar/gkab188) but ToPif1 does not neither form dimer in absence of ssDNA.

Therefore it is unlikely that DdPif1 could form dimers by domain swapping. However, this possibility is interesting from a structural point of view, and could occur in certain circumstances which have to be defined. That’s why we mentioned it in the manuscript.